# Research on Design and Performance of Self-Compacting Cement Emulsified Bitumen Mixture (CEBM)

**DOI:** 10.3390/ma15144840

**Published:** 2022-07-12

**Authors:** Jinming Yi, Jianlin Feng, Yuanyuan Li, Tao Bai, Anqi Chen, Yangming Gao, Fan Wu, Shaopeng Wu, Quantao Liu, Chuangmin Li

**Affiliations:** 1School of Civil Engineering and Architecture, Wuhan Institute of Technology, Wuhan 430205, China; jinmingyi_2005@163.com (J.Y.); baigs08@wit.edu.cn (T.B.); 22004010123@stu.wit.edu.cn (F.W.); 2Poly Changda Engineering Co., Ltd., Guangzhou 510062, China; 3State Key Laboratory of Silicate Materials for Architectures, Wuhan University of Technology, Wuhan 430070, China; angelchen@seu.edu.cn (A.C.); wusp@whut.edu.cn (S.W.); liuqt@whut.edu.cn (Q.L.); 4Faculty of Civil Engineering & Geosciences, Delft University of Technology, Stevinweg 1, 2628 CN Delft, The Netherlands; y.gao-3@tudelft.nl; 5School of Traffic and Transportation Engineering, Changsha University of Science and Technology, Changsha 410114, China; lichuangmin@csust.edu.cn

**Keywords:** bitumen/cement composite mixture, strength formation mechanism, early-strength, self-compacted, mixture performance

## Abstract

To meet the needs of the road industry for maintenance operations, a new cement emulsified bitumen mixture (CEBM) with early-strength, self-compacting, and room-temperature construction characteristics was designed. The strength formation mechanism of CEBM was revealed with a scanning electron microscope (SEM) and the surface free energy (SFE) theory. The mechanical properties and road performance of the CEBM were investigated extensively. The results show that before the demulsification of emulsified bitumen, the SFE of the bitumen–aggregate–water three-phase system was reduced due to the replacement of the bitumen–aggregate interface with water. The adhesion work between the emulsified bitumen and the aggregate is negative, which means the adhesion between the emulsified bitumen and the aggregate will not occur spontaneously due to the existence of water. The liquid emulsified bitumen improves the workability of the mixture and ensures that the mixture can be evenly mixed and self-compacted. After demulsification, the work of adhesion between the residual bitumen and the aggregate is positive, which means residual bitumen and aggregate can bond spontaneously. In addition, the hydration products of cement and aggregate form a skeleton, and the emulsified bitumen film wraps and bonds the cement and aggregate together, creating strength. The emulsified bitumen, cement content, and curing conditions have significant effects on the stability of CEBM. The recommended dosage of emulsified bitumen and cement is 8% and 8–10%, respectively. This material integrates the hardening effect of cement and the viscoelastic performance of bitumen and has good workability, mechanical properties, and road performance. Therefore, the CEBM is technically feasible for application to bitumen pavement.

## 1. Introduction

Due to the increase in the traffic volume and heavy load of the service process of asphalt pavement [1], a series of distresses such as high-temperature rutting, low-temperature cracking, fatigue cracking, and water damage gradually appear in the asphalt pavement, which significantly reduces the service level of asphalt pavement [2,3]. In addition, the mileage of the roads needing maintenance also increases rapidly year by year. The pothole and grooves of asphalt pavement are the main distressed types of asphalt pavement. Water damage, looseness, spalling, cracks, and other distresses of asphalt pavement may evolve into potholes and grooves of asphalt pavement as well [4,5]. These damages will significantly reduce the service level of asphalt pavement and affect the driving comfort and safety of asphalt pavement [6]. If they are not repaired in time, these distresses will develop rapidly under the comprehensive actions of traffic load and water, resulting in an increase in maintenance costs and seriously endangering driving safety [7,8]. Therefore, to meet the needs of the road industry for maintenance operations, it is of great significance to develop fast-setting and environment-friendly road maintenance materials.

The traditional repair materials for the potholes in asphalt pavement include hot-mix asphalt mixture (HMA) and cold-mix asphalt mixture (CMA) [9,10]. HMA has a great road performance and long life. However, the aggregate and bitumen need to be heated during the construction of asphalt pavement in processes such as hot mixing, hot paving, and hot compaction. After paving, asphalt pavement needs to be rolled by compaction machines, which is not only extremely inconvenient but also entails high labor costs and a slow construction speed. Therefore, the construction and production of HMA requires a lot of fuel and results in the emission of a amounts of greenhouse gases [11,12]. The fuel cost accounts for 15% of the total cost of HMA [13,14]. In addition, the high temperature will accelerate the release rate of volatile organic compounds in the asphalt mixture [15], which causes a great deal of harm to the environment and construction workers. On the other hand, the CMA [16,17] needs only simple mixing techniques. It requires the heating of the aggregate and bitumen during construction. However, it still needs to be compacted after paving, and the strength formation rate of CMA is slow, so the road performance of CMA is not as good as that of HMA. Based on the above analysis, the HMA has performance advantages, and the CMA has construction technology advantages, but they also have disadvantages. The cold-mixed cement/emulsified asphalt mixture can repair the potholes of asphalt pavement quickly and conveniently, which has both the advantages of the rigidity of cement concrete and the flexibility of asphalt mixture [18,19]. Compared with ordinary asphalt mixture, the cement/emulsified asphalt mixture can save a lot of energy, which is good for energy conservation and emission reduction [20]. However, the cement/emulsified asphalt mixture requires a certain amount of time to demulsify and hydrate. It also needs time for curing, which will prolong the construction period and affect traffic [21]; moreover, the fatigue and water damage resistance of cement/emulsified asphalt mixture is not good [22,23], which also limits the wide application of cement/emulsified asphalt mixture [24]. Furthermore, the above mixtures are not self-compacting, and all of them are required to be compacted after the paving [25], which virtually makes the operation and construction harder.

Many scholars have studied the strength formation mechanism of cement emulsified asphalt mixture. Anmar [26,27,28] studied the influence of emulsified asphalt and cement content on binary blended cement filler (BBCF), and the results showed that when the emulsified asphalt content was 8% and the cement content increased from 0 to 4%, the indirect tensile strength first increased and then decreased. When the cement content is constant at 3% and the emulsified asphalt content increases from 6% to 9%, the indirect tensile strength, compressive strength, and elastic modulus first increase and then decrease. The depth of the BBCF rutting test is about 1–2 mm. Mechanism research shows that the strength of cement emulsified asphalt mainly comes from two parts: hydration products formed by the hydration of cement and the demulsification of emulsified asphalt [23,26]. With the increase in the cement dosage, the number of hydration products increases correspondingly, especially in the immersion Marshall test, where the samples are tested in a water bath environment to ensure that the cement particles have enough water for hydration. Therefore, the Marshall stability value of the repair material increases with the increase of the cement dosage [28]. When the amount of emulsified asphalt is constant, with the increase in the cement content, the compressive strength of the repair material sample increases [29]. There are three main reasons: the increase in cement dosage is equivalent to the decrease in the water-binder ratio, so it has a positive effect on the strength of repair the materials [30]; the quantity of cement hydration products increases with the increase of cement dosage, so the strength of the repair materials increases accordingly [31]; the process of increasing the cement consumption consumes more free water and increases the demulsification process of emulsified asphalt, which is more conducive to the formation of the spatial network structure of emulsified asphalt-hydration products, thus providing strength for the repair materials [32]. However, the early strength development in the cement emulsified asphalt mixture presently studied is relatively slow, and it is not suitable for the roads that need to be repaired quickly, on the spot, and opened to traffic in a short time, such as earthquake-resistant roads, military roads, etc. Particularly, the repaired pavement is damaged again under the coupling action of rain and vehicle loads, which seriously affects the road capacity [33]. In addition, there is a lack of systematic research on cement emulsified asphalt mixture.

To compensate for the disadvantages of the above conventional materials in terms of construction technology, economy, environmental protection, and road performance, the use of a new bitumen/cement composite stabilized mixture with early-strength, self-compacting, and room-temperature construction characteristics (CEBM) for road maintenance is proposed, which may have adequate applications for repairing potholes in the asphalt pavement quickly and conveniently. Using sulphoaluminate fast-hardening cement and emulsified bitumen as the composite binders for CEBM can not only achieve the rigidity [34] enabled by cement but also meet the load-bearing capacity and improve high-temperature anti-rutting. The bitumen also has the flexibility [35] to meet the requirements of low-temperature cracking resistance and medium-temperature fatigue resistance [36]. It can be mixed under room temperature conditions to fill and repair potholes in asphalt pavement. The aggregate gradation is determined according to Gussasphalt concrete, commonly used for steel bridge deck pavement, which has a large ratio of bituminous material [37]. This mixture is self-compacting [38], so it is no need to compact during the construction [39]. At the same time, sulfoaluminate fast-hardening cement can quickly develop strength [40], Which is beneficial to quickly repair the road.

The surface micromorphology and interface energy of CEBM were studied through scanning electron microscopy (SEM) and surface energy theory, which can be used to reveal the strength formation mechanism of CEBM. The influences of the emulsified bitumen dosage, cement dosage, curing time, and other factors on the strength of CEBM were investigated to determine the design and preparation parameters of CEBM. In addition, the road performance properties, such as high-temperature rutting, low-temperature cracking, and water damage resistances of CEBM are also studied.

## 2. Materials and Experimental Methods

### 2.1. Materials

#### 2.1.1. Emulsified Bitumen

The emulsified bitumen that was used was cationic modified emulsified bitumen (BCR). BCR is prepared by emulsifying SBS modified bitumen. To prevent the segregation of the BCR, it should be stirred evenly before using and testing. The indicator test results are shown in Table 1.

#### 2.1.2. Cement

Cement that was used was sulphoaluminate fast-hardening and high-strength cement (SFHC). For comparison, the specific indicators of the SFHC and commonly used Portland cement (PO_42.5_) silicate cement are shown in Table 2. The technical performance of the SFHC met the technical requirements of Specification for Design of Highway Cement Concrete Pavement (JTG D40-2011).

#### 2.1.3. Aggregates and Fillers

The coarse and fine aggregates that were used comprised crushed limestone. The particle size of the coarse aggregate is ≥2.36 mm, while that of the fine aggregate is 0.075–2.36 mm. The filler that was used was the mineral powder ground from the limestone; the results of its technical properties are shown in Table 3, Table 4 and Table 5.

### 2.2. Mixture Gradation

Gussasphalt concrete is a kind of asphalt mixture with high asphalt content, high mineral powder content, and a void ratio of less than 1%, which is mixed at high temperature (220~260 °C) and paved by the fluidity of the mixture itself without rolling. The CEBM has great flowability with a self-compaction property after paving, it has high bitumen and high mineral powder content, and the gradation of CEBM is similar to that of Gussasphalt concrete. The gradation range of Gussasphalt concretes varies among different countries. The gradation of CEBM was selected from the range intersection of Gussasphalt concrete in China, Germany, and the EU [41]. The aggregate composite gradation is shown in Figure 1. Table 6 is the proportion of each mineral aggregate. The cement is used to replace the part of the mineral powder in equal volume, and the sum of cement and mineral powder accounts for 24% the weight of the mineral aggregates. The emulsified bitumen content is measured in the weight of aggregates, and its proportion is calculated with the residue bitumen after demulsification.

### 2.3. Mechanism Characterization Test

#### 2.3.1. Surface Micro-Morphology Testing (SEM)

A Scanning Electron Microscope (SEM, JSW-5510LV) was applied to investigate the surface micro-morphology of CEBM, and the test voltage was 15 mv. The sample was collected from the middle of the CEBM specimen, cured for 24 h, and dried in an oven at 45 °C. Then, a gold spraying treatment was conducted on the surface of the sample, to ensure the sample conduction. The sample was placed on the sample table of SEM for detection.

#### 2.3.2. Surface Free Energy Measurements

The surface free energy (SFE) of the residual bitumen from the emulsified bitumen, 70# base bitumen, styrene-butadiene-styrene (SBS) modified bitumen, limestone aggregate, and cement concrete were tested. For the preparation of the bitumen sample, the bitumen was dropped on a glass slide and placed horizontally in an oven at 150 ± 5 °C for 3 min to level the bitumen surface. Then, it was naturally cooled in a dust-free and dry environment for 6 h. The heated and liquified bitumen were spread onto a dried heat-resistant microscope slide to form a homogeneous film. The bitumen specimen is shown in Figure 2a. For the aggregate and cement specimen, limestone rock and 7 d cured cement block were cut into 50 mm × 50 mm × 5 mm and a polishing treatment of their surfaces was performed to obtain a smooth specimen. The specimen is shown in Figure 2b.

The bitumen (film), limestone, and cement concrete were tested using three liquids with known SFE parameters, namely distilled water, ethylene glycol, and glycerol, respectively. The distilled water, ethylene glycol, and glycerol were used to measure the contact angle; the surface free energy parameters of these three kinds of liquid are shown in Table 7. The pendant-drop method was applied and the contact angle was measured with a contact angle tester, as shown in Figure 3, the test temperature was 25 °C. The experiment was carried out using an SDC-100 contact angle device that was sourced from Dongguan Dingsheng Precision Instrument Co., Ltd.

### 2.4. Manufacturing of CEBM

CEBM adopts a Gussasphalt concrete gradation, based on a C/EB formula system with high asphalt content, high mineral powder (including cement) content, and a small amount of water, which grants it great flowability. Using its great flowability “pouring, leveling, and compacting”, it can form a uniform pavement with high density and low voids without rolling.

Firstly, Gussasphalt concrete gradation was adopted to ensure that the air voids of the molding mixture sample were less than 1%. The emulsified bitumen, cement, water, aggregate, and mineral powder were measured according to the proportion ratio of CEBM. In addition, the water-cement ratio was determined to be 0.5:1 [18]. The Additional water quantity of CEBM was then calculated according to Equation (1).
(1)$$C_{W}=R_{w/c}C_{C}-\left(1-C_{Br}\right)C_{B}$$
where *C_W_* is the additional water content in CEBM, *R_w/c_* is the water-cement ratio, *C_C_* is the cement content in CEBM, *C_B_* is the emulsified bitumen content in CEBM, and *C_Br_* is the evaporation residue of the emulsified bitumen.

Then, the aggregates were mixed in dry conditions for an even mixture; the designed amount of water was sprayed to the uniformly mixed aggregate, it was stirred quickly and evenly, thus granting the mixture the desired flowability, and flowability experiment’s outflow time was less than 20 s (JTG/T3364-02—2019). The mixture was conditioned at room temperature (25 °C) for curing for more than 24 h. The molding process and specimens of CEBM are shown in Figure 4. The Volume parameters of CEBM are shown in Table 8.

### 2.5. Experimental Methods

#### 2.5.1. Mechanical Property Test of CEBM

The Marshall stability of the Marshall specimen with different cement and emulsified bitumen contents and the curing time were detected to investigate the influential parameters of the mechanical property test of CEBM. Each group had 8 specimens, 4 of which were conditional and 4 were non-conditional. After curing them at 25 °C for 6 h, 12 h, and 24 h, respectively, the Marshall stability of CEBM was tested under the drying and bath water conditions of 25 °C and 60 °C, respectively.

#### 2.5.2. Wheel Track Test of CEBM

The size of the rutting specimen of CEBM was 300 mm × 300 mm × 50 mm, and the specimens were cured at 25 °C for 24 h and 72 h (Complete curing). Before the test, the samples were placed in the rutting instrument for more than 5 h to maintain a constant temperature, and the test temperature, wheel loading, and rate were 60 °C, 0.7 MPa, and 42 times/min, respectively.

#### 2.5.3. Low-Temperature Bending Test of CEBM

The Universal testing machine (UTM-100) was used for testing the low-temperature crack resistance of CEBM. The specimen size was 250 mm × 30 mm × 35 mm. Each group had 3 specimens. The test temperature was −10 °C, the span of the beam was 200 mm, and the loading rate was 50 mm/min. The schematic diagram is shown in Figure 5.

#### 2.5.4. Water Sensitivity Test of CEBM

The immersion Marshall stability test and the freeze-thaw splitting test were conducted according to the criteria of the “Standard Test Method of Bitumen and Bituminous Mixtures for Highway Engineering” JTG E20-2011. Each group had 8 specimens, 4 of which were conditional and 4 were non-conditional, which were used to test the water stability of CEBM with different cement and emulsified bitumen contents.

### 2.6. Test Flow Chart

The test flow chart is shown in Figure 6.

## 3. Experimental Results and Discussion

### 3.1. Analysis of Surface Morphology and Surface Energy

#### 3.1.1. Surface Micro-Morphology of CEBM

Cement and emulsified bitumen were used as the composite binders in the CEBM. A SEM was used to observe the micro-morphology of CEBM cured for 24 h. The 1500×, 3000×, 5000×, and 8000× magnified figures of the sample are shown in Figure 6. The hydration products of cement mainly include acicular and reticulated hydrated calcium silicate (C-S-H), flaky calcium hydroxide (C-H), and columnar ettringite (C-A-S-H) [43,44]. In Figure 7a,b, there are many micropores in the CEBM, which are the air voids where water evaporates from the demulsification of emulsified bitumen. In Figure 7c, the flake calcium hydroxide is well wrapped by the demulsified bitumen. From Figure 7c,d, the hydration products of cement that can be observed are uniformly distributed in the demulsified bitumen. And the circular hydrated calcium silicate, columnar ettringite, and flaky calcium hydroxide can be clearly observed in the CEBM [18,21]. In Figure 7d, the emulsified bitumen forms a film after demulsification and wraps the surface of the aggregate, mineral powder, and cement concrete, and the hydration products of cement are also evenly distributed in the CEBM. The hydration products can pierce the bitumen film and smooth areas, and form bonds with other hydration products or the aggregate surface. The bitumen and cement composite bind the aggregate and mineral power together [45].

On the one hand, water and liquid emulsified bitumen can significantly increase the flowability of the CEBM, so it can mix at room temperature and has the advantage of self-compaction. On the other hand, the cementitious phase in the CEBM was dispersed within the emulsified bitumen. The cement hydration consumed a portion of the water that occupies the micro air void spaces between the emulsified bitumen and aggregate, which had a stiffening effect on emulsified bitumen [43,46]. The hydration products of the aggregate and cement, the skeleton function of the cement, and the encapsulation and adhesion function of the emulsified bitumen complement each other, and together with the hydration products of cement interweave with demulsified bitumen to enhance the overall stability and form the strength of CEBM.

#### 3.1.2. Surface Free Energy Analysis of CEBM

In the SFE theory, the surface tension of matter is composed of the Lifshitz-Van Der Waals interaction ($\gamma^{d}$) and the Lewis acid-base interaction ($\gamma^{+/-}$) [47,48]. The contact angles of distilled water, glycol, and glycerol titration on the surfaces of bitumen, aggregate, and cement blocks were measured to calculate their surface energy. The surface energy parameters of the bitumen and aggregate were calculated by Young’s Equations (2)–(4) [47,49]. Via substitution into Young’s equation, the three unknowns ($\gamma^{d},\gamma^{+},\;\gamma^{-}$) were solved simultaneously. The results are shown in Table 9.
(2)$$\gamma_{L}\frac{1+\cos\theta }{2}=\sqrt{\gamma_{S}^{d}\gamma_{L}^{d}}+\sqrt{\gamma_{S}^{+}\gamma_{L}^{-}}+\sqrt{\gamma_{S}^{-}\gamma_{L}^{+}}$$
(3)$$\gamma_{L}=\gamma_{}^{d}+\gamma_{L}^{p}$$
(4)$$\gamma_{L}^{p}=2\sqrt{\gamma_{L}^{+}\gamma_{L}^{-}}$$
where $\gamma_{S}^{d},\gamma_{S}^{+},\gamma_{S}^{-}$ represent the dispersion component, Lewis acid number, and Lewis base number of the tested solid, respectively; $\gamma_{L},\gamma_{L}^{d},\gamma_{L}^{+},\gamma_{L}^{-}$ express the surface free energy, dispersion component, Lewis acid number, and Lewis base number of the test liquor, respectively.

Adhesion work refers to the work performed when separating the two phases that contact (adhere to) each other on two new surfaces. The adhesion work >0 indicates that the adhesion process can proceed spontaneously. The adhesion process between the residual bitumen of the emulsified bitumen and aggregate can be explained by SFE theory, and SFE can calculate the adhesion work between the bitumen and aggregate. The cohesive work of the same (single-phase) substance was computed using Equation (5); the interface between the two substances (two-phase) can be calculated according to Equation (6) [48,50]. However, The volatilization of water and hydration of cement are the main reasons for the rapid strength formation of CEBM. Therefore, it is necessary to study the adhesion work between the emulsified bitumen, the cement block, and the aggregate before and after the demulsification of the emulsified bitumen. According to the surface energy theory, before bitumen demulsification, water, bitumen, and aggregate coexist, and the adhesion work W_bsw_ of the three-phase system of water, bitumen, and aggregate is calculated by the Equation (7) [51].
(5)$$\gamma_{\mathrm{ii}}=2\gamma_{i}=2\left(\gamma_{i}^{d}+\gamma_{i}^{p}\right)$$
(6)$$\gamma_{\mathrm{ij}}=2(\sqrt{\gamma_{i}^{d}\gamma_{j}^{d}}+\sqrt{\gamma_{i}^{+}\gamma_{j}^{-}}+\sqrt{\gamma_{i}^{-}\gamma_{j}^{+}})$$
(7)$$W_{bsw}=\gamma_{bs}+\gamma_{ww}-\gamma_{bw}-\gamma_{sw}$$
where $\gamma_{ii}$ is the cohesive work of the same substance; $\gamma_{ij\;}$is the interface energy of two substances; *W_bsw_* is the adhesion work of the water, bitumen, and aggregate three-phase system before demulsification; $\gamma_{\mathrm{bs}},\gamma_{\mathrm{bw}},\gamma_{\mathrm{sw}}$ represent the adhesion work between the bitumen and aggregate, bitumen and water, and aggregate and water, respectively; $\gamma_{\mathrm{ww}}$ is the cohesive force between water and water; and $\gamma_{i}^{d},\gamma_{i}^{+},\gamma_{i}^{-}$ represent the dispersion component, Lewis acid, and base number (*i* and *j* are the *b*, *s*, and *w*; the *b*, *s*, and *w* represent bitumen, aggregate, and water, respectively).

The adhesion work of bitumen towards the aggregate and cement mortar block was calculated by substituting the parameters in Table 9. The adhesion work >0 indicates that the adhesion process can proceed spontaneously. The results are shown in Figure 8. Before demulsification, the SFE of the bitumen–aggregate–water three-phase system was reduced due to the existence of water in the bitumen–aggregate interface. The adhesion work between the emulsified bitumen and aggregate is negative, and the adhesion between emulsified bitumen and aggregate may not happen spontaneously due to the existence of water. Therefore, the liquid emulsified bitumen can improve the workability of the mixture and ensures that the mixture can be evenly mixed and self-compacted. The adhesion work of the emulsified bitumen with the limestone aggregate and cement mortar before demulsification is −129.97 mJ/m^2^ and −108.80 mJ/m^2^, that is, the surface free energy changes are 129.97 mJ/m^2^ and 108.80 mJ/m^2^, which shows that the adhesion of the emulsified bitumen with limestone aggregate and cement mortar cannot occur without other physicochemical effects.

After demulsification, the adhesion work between the residual bitumen and aggregate is positive, and the residual bitumen and aggregate can bond spontaneously. The demulsification film of bitumen further wraps and adheres to the aggregate, and establishes a spatial network structure in the mixture, thus forming strength. After the demulsification of emulsified bitumen, the adhesion work between 70# bitumen and SBS modified bitumen with the aggregate is positive, which means the change of surface free energy is negative, indicating that their adhesion is spontaneous and can be bonded without external work. In addition, after the demulsification of the emulsified bitumen, the adhesion work of emulsified bitumen towards the limestone and cement mortar block is 75.09 mJ/m^2^ and 171.36 mJ/m^2^, respectively; the adhesion work of 70# bitumen towards the limestone and cement mortar block is 65.24 mJ/m^2^ and 144.61 mJ/m^2^, respectively; and the adhesion work of SBS modified bitumen towards the limestone and cement mortar block is 74.24 mJ/m^2^ and 122.73 mJ/m^2^, respectively. The adhesion work of emulsified bitumen after demulsification towards limestone is 15.1% and 1.1% higher than that of 70# bitumen and SBS modified bitumen, and the adhesion work of emulsified bitumen after demulsification towards cement mortar is 18.5% and 39.6% higher than that of 70# bitumen and SBS modified bitumen. The results show that the adhesion work of emulsified bitumen after demulsification with limestone and cement mortar is higher than that of 70# bitumen and SBS modified bitumen, which can ensure that the CEBM possesses good water damage resistance.

### 3.2. Influence Factors of Mechanical Performance of CEBM

#### 3.2.1. Influence of Cement and Emulsified Bitumen Content on Mechanical Performance of CEBM

The Marshall specimens are prepared with 8% and 10% emulsified bitumen and 8%, 10%, and 12% sulphoaluminate cement. The flowability of CEBM with different cement and emulsified bitumen contents is shown in Table 10, and the test results of its Marshall stability are shown in Figure 9. In Figure 9, the Marshall stability of CEBM increases with the increase of the emulsified bitumen (8–12%) and cement (8–12%) contents; when the cement content is 12%, no large difference is observed for the Marshall stabilities of CEBM with 8% and 10% emulsified bitumen content. The CEBM with 12% cement and 8% emulsified bitumen has the largest Marshall stability of 10.88 kN, while with 8% cement and 8% emulsified bitumen, it has the lowest stability (8.26 kN). However, it still satisfies the requirement of the Marshall stability of HMA (≥8 kN).

The hydration of cement makes use of the free water produced by the demulsification of emulsified bitumen, which accelerates the demulsification of bitumen [16,44]. Emulsified bitumen residues and cement hydration products are the composite binders of the CEBM, which explains why the strength of the mixture increases gradually with the increase of the cement content. For emulsified bitumen, the free water produced by demulsification not only promotes the hydration reaction of cement but also enables the mixture to have great flowability and self-compacting characteristics [45,47]. With the increase of the emulsified bitumen content, the flowability of the mixture is enhanced. However, with the further increase of the emulsified bitumen content, both the free bitumen and free water content will increase [52,53] such that the 12% cement content may increase the free water content and free bitumen content, and the free water will produce the air void in the CEBM after its evaporation; therefore, the free bitumen act as the lubricant between the aggregates and cement concrete, thereby reducing the mechanical performance of CEBM.

#### 3.2.2. Curing Time Effect on the Mechanical Performance of CEBM

The strengths of the CEBM with 8% emulsified bitumen and 8% and 10% cement are tested after 6 h, 12 h, and 24 h curing, respectively, and the test results are shown in Figure 10. Figure 10 shows that the Marshall stability of CEBM increases continuously with the increase of the curing time (0–24 h). In detail, the Marshall stability of CEBM with 8% BCR and 10% cement after 24 h curing is 68.1% and 167.7% higher than that of CEBM after 12 h and 6 h curing. When the curing time is 6 h, the Marshall stability of the CEBM with 8% emulsified bitumen and 10% cement is 3.44 kN, which satisfies the requirement of the Marshall stability CMA (≥3 kN). After 24 h curing, the Marshall stability of the CEBM meets the requirements of HMA (≥8 kN); therefore, it has a good early-strength property.

#### 3.2.3. Test Conditions Effect on Mechanical Performance of CEBM

The mechanical performance of CEBM was tested under different test conditions, such as 25 °C drying, a 25 °C water bath, and a 60 °C water bath, to investigate the test condition’s effect on the mechanical performance of CEBM. The results are similar to Figure 11. From Figure 11, the CEBM shows the highest Marshall stability under the 25 °C drying. After the 25 °C water bath, the stability decreased slightly, and the stability of the CEBM with 8% BCR + 8% cement and 8% BCR + 10% cement cured for 6 h decreased by 7.9% and 3.1%, respectively, while the stability of the CEBM with 8% BCR + 8% cement and 8% BCR + 10% cement cured for 24 h decreased by 2.4% and 9.6%, respectively. After the 60 °C water bath, the stability of CEBM decreases more obviously. The changing trend of the stability of CEBM under different test conditions is consistent.

### 3.3. Mixture Performance Test of CEBM

#### 3.3.1. High Temperature Stability of CEBM

The rutting resistance of the mixture cured for 24 h and 72 h was studied by a wheel track test. The results are shown in Figure 12 and Figure 13. From Figure 12, with the increase of the loading cycles, the cumulative deformation (rutting depth) of the mixture gradually increases, and the rutting depth growth rate of the CEBM cured for 72 h is lower than that of the CEBM cured for 24 h. Under conditions with the same number of loading cycles, the rutting depth of the CEBM with 72 h of curing time is much lower than that of the CEBM with 24 h of curing time. When the loading time is 45 min and 60 min, the rutting depth of the CEBM cured for 72 h is 0.167 mm and 0.173 mm, respectively, and for 24 h is 0.272 mm and 0.308 mm, respectively. The rutting depth of CEBM cured for 72 h is 62.87% and 78.03% less than that of maintenance for 24 h. The maximum rutting depth of CEBM is 0.3 mm, which is far less than the existing 1–2 mm rutting depth of the cement emulsified bitumen [27]. From Figure 13, after curing for 24 h, the dynamic stability (DS) of CEBM is 18,333 times/mm, and for 72 h, it is 63,000 times/mm, which is much higher than the requirement of HMA, indicating that due to the high strength offered by the cement, the high-temperature stability of CEBM is very good.

#### 3.3.2. Low-Temperature Crack Resistance of CEBM

The low-temperature bending test was conducted to evaluate the low-temperature crack resistance of CEBM. The results are shown in Figure 14 and Table 11. The maximum bending tensile strain of ordinary cement emulsified bitumen mixture trabecula is about 2100 με, and the bending stiffness modulus is about 1800 MPa [28]. According to Figure 14, the maximum failure loading of CEBM is 0.46 kN, and the corresponding mid-span deflection is 0.52 mm. From Table 11, the bending tensile strength of the trabeculae is 4.28 MPa. Compared with ordinary cement emulsified bitumen mixture, the maximum bending tensile strain of CEBM increases by 19.19%., and the bending stiffness modulus decreases by 4.98%. The Low-temperature performance has been improved to some extent. Although there is no technical requirement for the failure strain in the low-temperature bending test of CMA, the maximum flexural tensile strain of CEBM still meets the requirement of China’s criterion (JTG D50-2017) that the flexural strain of the HMA should be greater than 2000με. This performance is improvement due to the existence of the emulsified bitumen, which can provide flexibility for CEBM and improve the Low-temperature crack resistance of CEBM.

#### 3.3.3. Water sensitivity of CEBM

(1)Immersion Marshall test

The results of the water immersion Marshall test of CEBM at 25 °C and 60 °C are shown in Table 12. According to Table 12, the Marshall stabilities of CEBM both before and after water immersion are higher than 8 kN (the HMA requirement), and the residual Marshall stability of CEBM after water immersion increases at first and then decreases with the increase of the cement content. The water-immersed residual stability (IRS) of ordinary cement emulsified bitumen mixture is 85.3% [28]. The IRS of CEBM is shown in Figure 15. The 25 °C water immersion-conditioned CEBM with 8% BCR and 8% cement has the smallest IRS value of 92.4%, and the 60 °C IRS of CEBM with 8% BCR and 10% cement has the smallest IRS of 130.3%. The 25 °C IRS of CEBM increases with the increase of cement content, while the 60 °C IRS of CEBM initially increases and then decreases with the increase of the cement content, and reaches the maximum value at 10% cement content. All of the IRS values of CEBM are greater than ordinary cement emulsified bitumen mixture, which is obviously better than the requirements of higher than 80%.

(2)Freeze-thaw splitting test

The results of the freeze-thaw indirect tensile strength test are shown in Figure 16. According to Figure 16, the freeze-thaw indirect tensile strength ratio (TSR) of the CEBM with 10% BCR and 8% cement content is the smallest value, which is 83.3%, and the TSR value of the CEBM with 8% BCR and 12% cement content is the highest, which is 105.3%. The TSR of the mixture increases with the increase of the cement content. Since the splitting strength of the conditional group in the freeze-thaw splitting test is about 0.4 MPa, the TSR increases with the decrease of the splitting strength of the unconditional group. This explains why with the increase in the cement content, although the TSR increases, its splitting strength also decreases. In addition, all of the TSR values of the above CEBMs meet the requirements of higher than 75%. Therefore, the CEBM has good water stability.

## 4. Conclusions

The SEM and surface energy theory of the CEBM were studied using modern testing technology, and the strength formation mechanism of CEBM was revealed. The influence of the emulsified bitumen, the cement dosage, and the curing time on the strength of CEBM was studied. In addition, the road performance of CEBM is evaluated by road performance tests. The following conclusions were obtained.

(1) Before demulsification, the SFE of the bitumen–aggregate–water three-phase system was reduced due to the replacement of the bitumen–aggregate interface with water. The adhesion work between the emulsified bitumen and aggregate is negative, and the adhesion between the emulsified bitumen and aggregate may not happen spontaneously due to the existence of water. Meanwhile, water exists in bitumen and aggregate, which improves the workability of CEBM and ensures its uniform mixing and self-compacting. After demulsification, the adhesion work between the residual bitumen and aggregate is positive, and the residual bitumen and aggregate can bond spontaneously. The free water produced by the demulsification of bitumen reacts with the cement, the hydration products of cement form a skeleton in aggregate, and the demulsified bitumen further encapsulates the aggregate and cement and bonds them together. The skeleton of the cement and the adhesion of bitumen complement each other, and establish a spatial network structure in the CEBM, thus forming high strength.

(2) The emulsified bitumen content, cement content, and curing conditions have significant effects on the mechanical stability of CEBM. When the cement content is 12% and the emulsified bitumen content is 8%, the CEBM has the maximum Marshall stability of 10.88 kN; when the cement content is 8% and the emulsified bitumen content is 8%, the CEBM has the maximum Marshall stability of 8.26 kN. All of these values are even higher than the requirement for the hot mix bitumen mixture (≥8 kN). In addition, when the curing time is 6 h, all the Marshall stabilities of CEBM can reach the stability requirement of CMA (≥3 kN).

(3) Due to the hardening effect of cement, the CEBM has an excellent rutting resistance at high temperatures, and the dynamic stability is 18,333 times/mm cured for 24 h. On the other hand, due to the viscoelasticity of bitumen, the maximum flexural-tensile strain at low temperature is 2503 με, which even meets the requirement of the flexural-tensile strain of hot mix bitumen mixture (≥2000 με). The water immersion residue is higher than 110% with 10% cement, and the TSR is higher than 85%, indicating the CEBM has good water stability.

(4) CEBM has good working and mechanical properties; therefore, it is technically feasible to use CEBM as a new material for road construction and maintenance. For comprehensive economic considerations, the recommended dosage of CEBM emulsified bitumen is 8%, and that of cement is 8–10%. The cement dosage can be determined according to the relevant engineering requirements.

## Figures and Tables

**Figure 1 materials-15-04840-f001:**
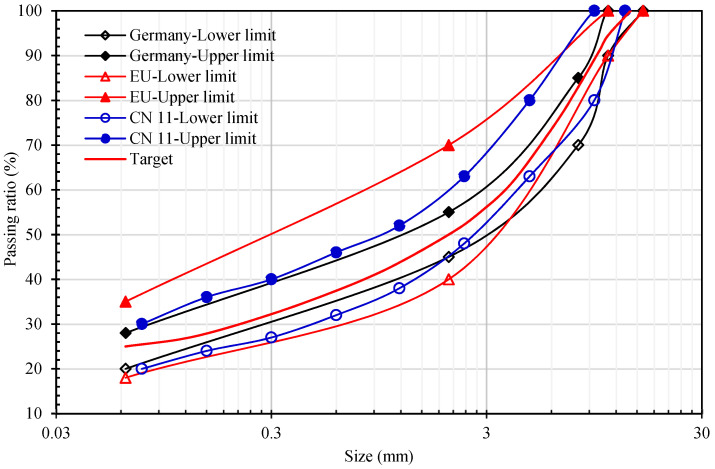
Gradation curve.

**Figure 2 materials-15-04840-f002:**
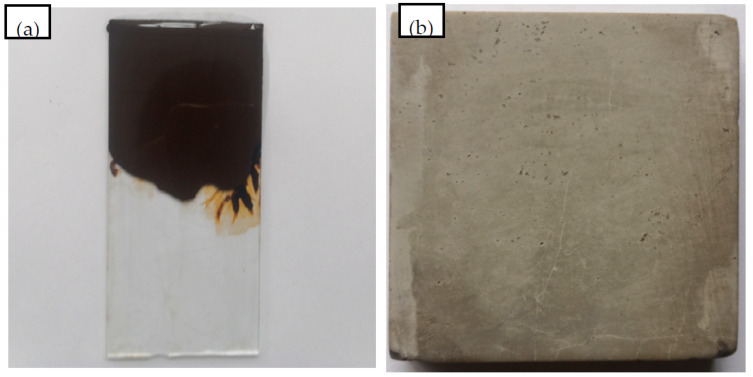
Contact angle test of bitumen and cement mortar. ((**a**) The bitumen specimen; (**b**) limestone rock specimen).

**Figure 3 materials-15-04840-f003:**
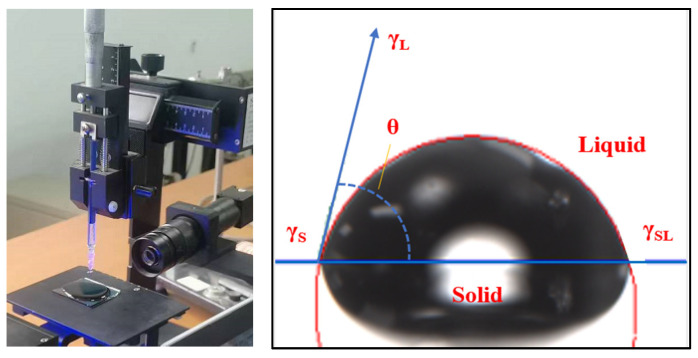
Diagram of contact angle test.

**Figure 4 materials-15-04840-f004:**
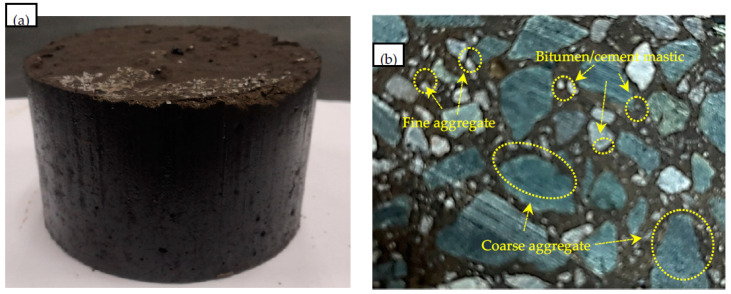
Manufacturing of CEBM ((**a**) Marshall specimen appearance; (**b**) apparent section of Marshall specimen).

**Figure 5 materials-15-04840-f005:**
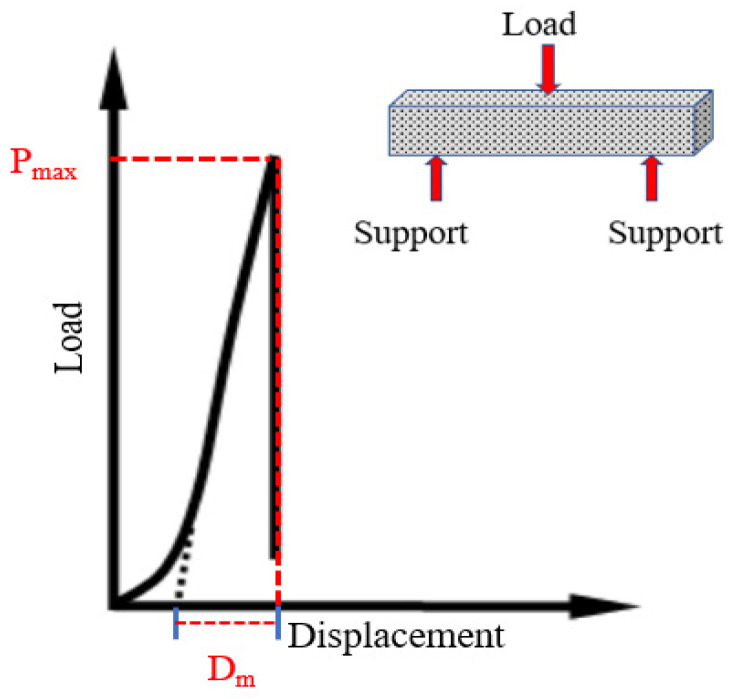
Three-point bending test [42].

**Figure 6 materials-15-04840-f006:**
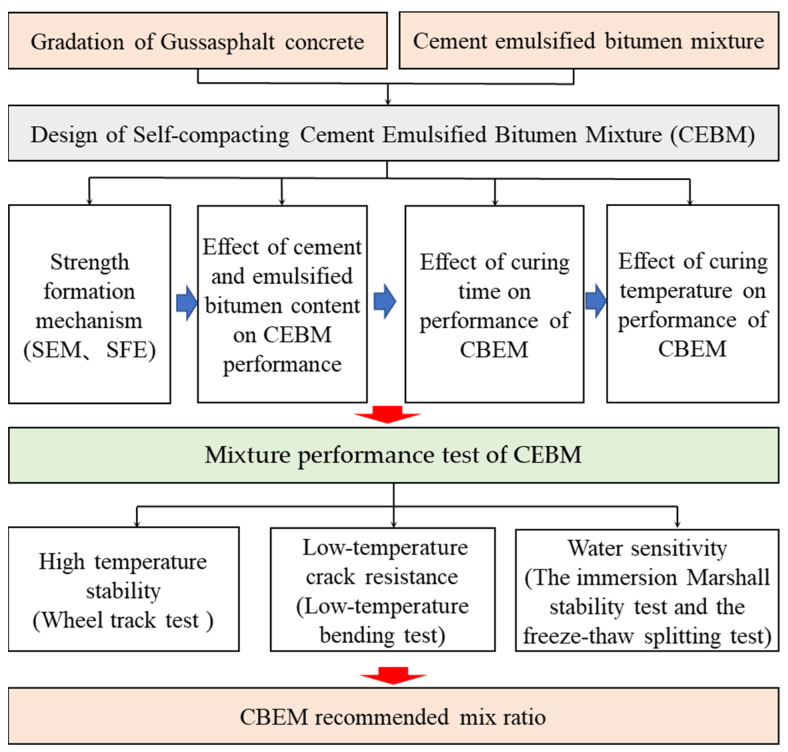
Test flow chart.

**Figure 7 materials-15-04840-f007:**
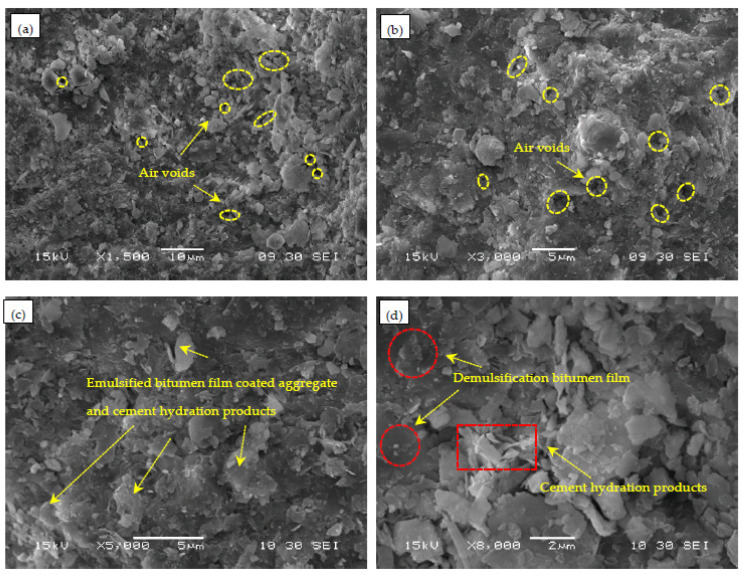
Microscopic morphology figures of CEBM ((**a**) 1500×; (**b**) 3000×; (**c**) 5000×; (**d**) 8000×).

**Figure 8 materials-15-04840-f008:**
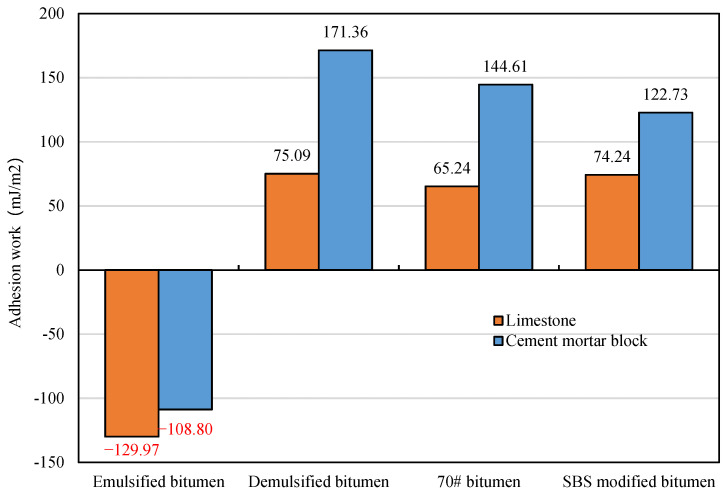
Bitumen-aggregate interface adhesion work.

**Figure 9 materials-15-04840-f009:**
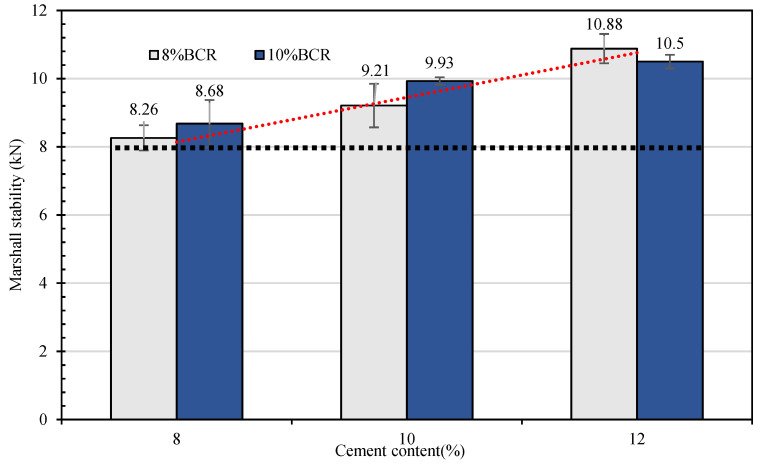
Marshall stability of CEBM with different cement and emulsified bitumen contents.

**Figure 10 materials-15-04840-f010:**
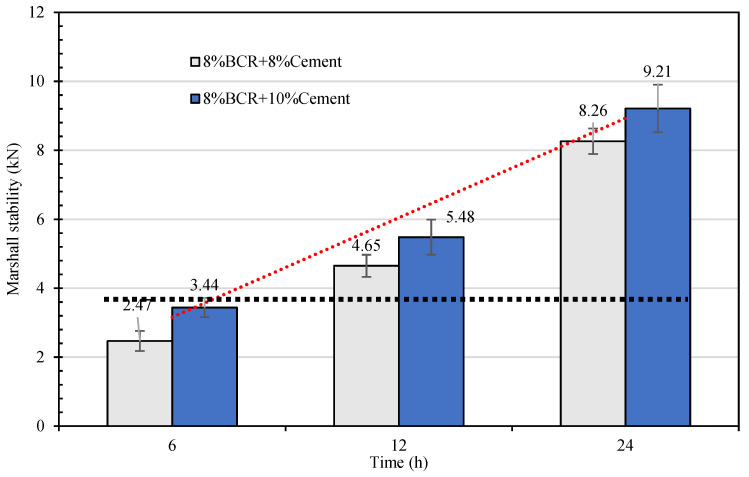
Marshall stability of CEBM after 6 h, 12 h, and 24 h curing times.

**Figure 11 materials-15-04840-f011:**
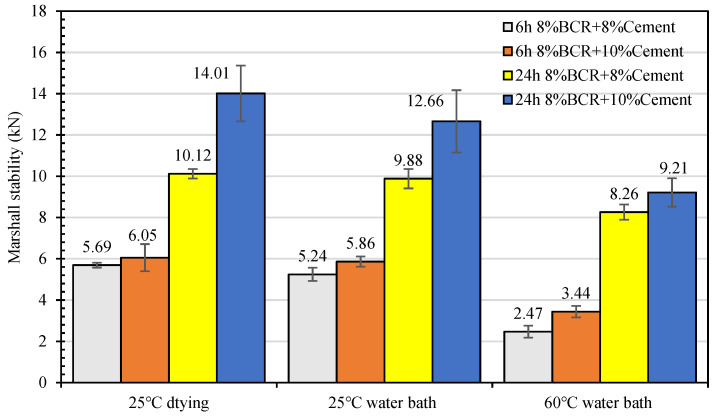
Marshall stability of CEBM under different test conditions.

**Figure 12 materials-15-04840-f012:**
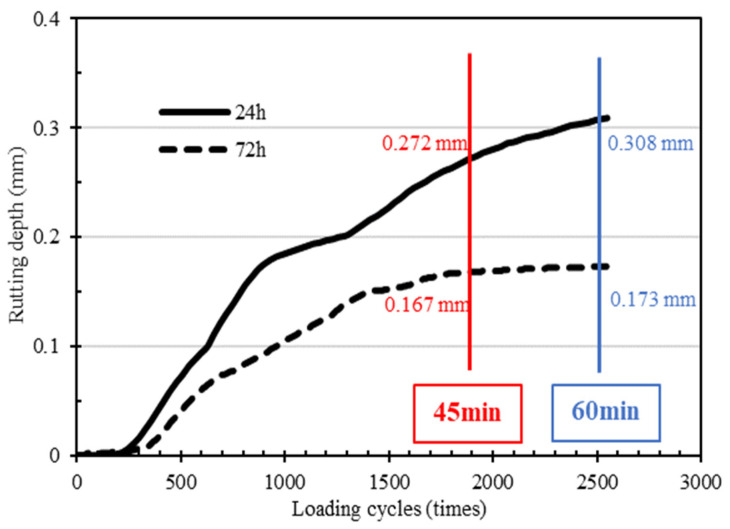
Rutting increasing curve of CEBM.

**Figure 13 materials-15-04840-f013:**
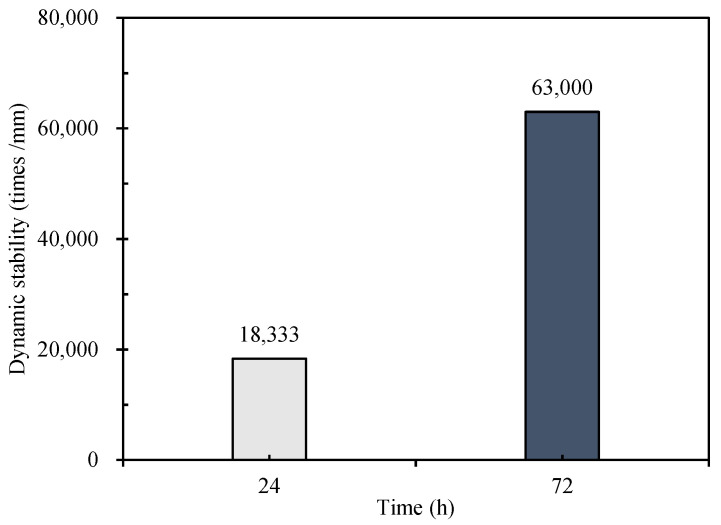
Dynamic stability of CEBM.

**Figure 14 materials-15-04840-f014:**
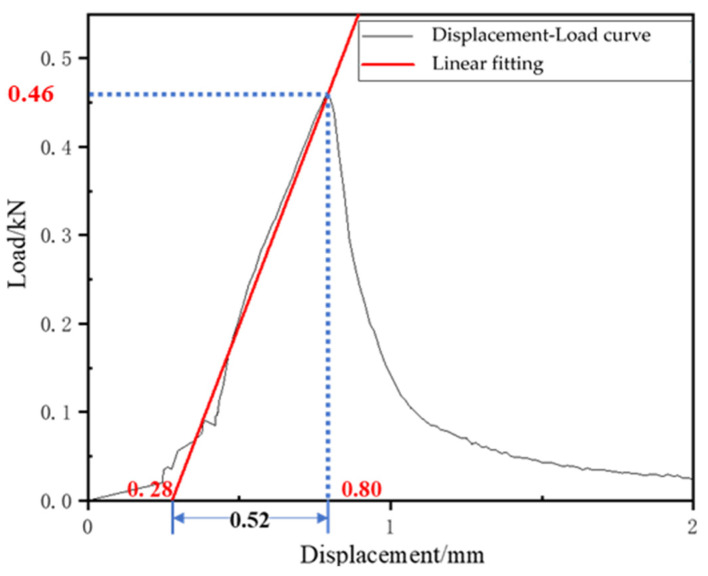
Loading-deformation curve of low-temperature bending test.

**Figure 15 materials-15-04840-f015:**
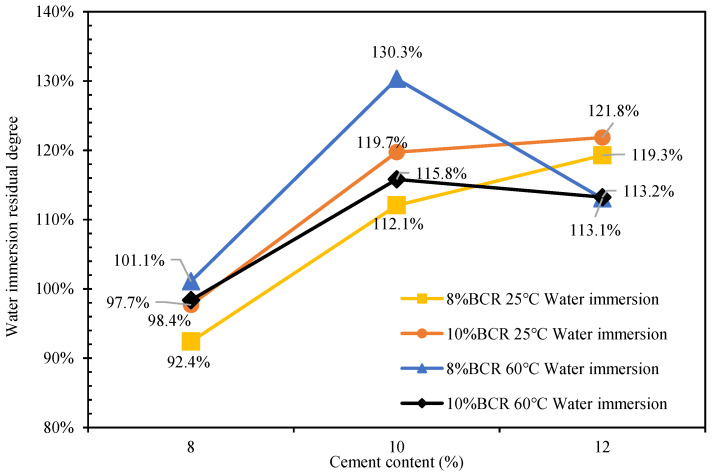
Water-immersed residual stability (IRS) of CEBM.

**Figure 16 materials-15-04840-f016:**
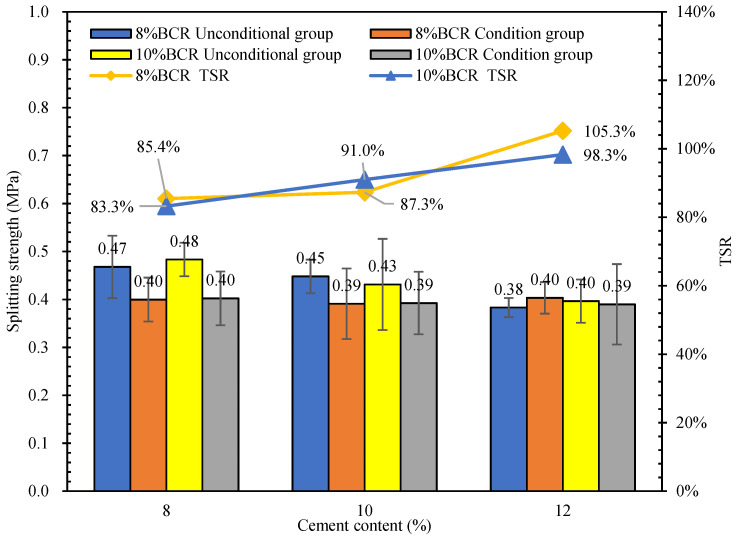
Splitting strength and freeze-thaw splitting ratio of CEBM.

**Table 1 materials-15-04840-t001:** Technical properties of emulsified bitumen.

Test Item		Unit	Results	Requirements	Experimental Method
Demulsification speed		–	Slow crack	Slow crack	JTG E20-2011 T0658
Particle charge		–	+	+	JTG E20-2011 T0653
Evaporation residual degree		%	58.6	≥55	JTG E20-2011 T0651
Penetration of evaporated residue bitumen (25 °C)		0.1 mm	56	45–150	JTG E20-2011 T0604
Ductility of evaporated residue bitumen (15 °C)		cm	45.5	≥40	JTG E20-2011 T0606
Softening point of evaporated residue bitumen		°C	49.4	–	JTG E20-2011 T0605
Normal temperature storage stability	1 d	%	0.47	≤1	JTG E20-2011 T0655
	5 d		2.17	≤5	

**Table 2 materials-15-04840-t002:** Technical properties of SFHC and PO_42.5_ silicate cement.

Cement	Specific Surface Area (m^2^/kg)	Setting Time (min)		Compressive Strength (MPa)			Flexural Strength (MPa)		
		Initial Setting	Final Setting	1 Day	3 Days	28 Days	1 Day	3 Days	28 Days
SFHC	431	15	31	30	41.2	52.1	3.2	4.1	7.3
PO_42.5_ silicate cement	396	175	235	8	27.5	49	1.3	5.5	8.0

**Table 3 materials-15-04840-t003:** Technical properties of coarse aggregate.

Parameters	Unit	Results	Requirements	Experimental Method
Stone crushing value	%	23.2	≤28	JTG E42-2005 T0316
Needle flake content	%	8.2	≤15	JTG E42-2005 T0312
Los Angeles wear value	%	15	≤28	JTG E42-2005 T0317
Water absorption	%	0.9	≤2.0	JTG E42-2005 T0308
Apparent specific gravity	–	2.667	≥2.6	JTG E42-2005 T0605

**Table 4 materials-15-04840-t004:** Technical properties of fine aggregate.

Parameters	Unit	Results	Requirements	Experimental Method
Sediment percentage	%	2	≤3	JTG E42-2005 T0335
Sand equivalent	%	63	≥60	JTG E42-2005 T0334
Angularity (flow time method)	s	45	≥30	JTG E42-2005 T0345
Apparent specific gravity	–	2.650	≥2.5	JTG E42-2005 T0328

**Table 5 materials-15-04840-t005:** Technical properties of filler.

Parameters		Unit	Results	Requirements	Experimental Method
Apparent specific gravity		–	2.612	≥2.5	JTG E42-2005 T0352
Particle size range (%)	<0.6	mm	100	100	JTG E42-2005 T0351
	<0.15	mm	92.4	90–100	
	<0.075	mm	86.3	75–100	
Plasticity coefficient		–	3.5	<4	JTG E42-2005 T0354
Hydrophilic coefficient		–	0.82	<1	JTG E42-2005 T0353

**Table 6 materials-15-04840-t006:** Grading and proportion of mineral materials.

Aggregate	10–15 mm	5–10 mm	3–5 mm	0–3 mm	Filler
Proportion	4%	12%	32%	28%	24%

**Table 7 materials-15-04840-t007:** SFE parameters of test liquid (25 °C, mJ/m^2^).

Reagent	SFE $\left(\gamma_{L}\right)$	Dispersion Component $\left(\gamma_{L}^{d}\right)$	Polarity Component $\left(\gamma_{L}^{p}\right)$	SFE Acidity Pa-rameter $\left(\gamma_{L}^{+}\right)$	SFE Alkalinity Parameter $\left(\gamma_{L}^{-}\right)$
Distilled water	72.8	21.8	51.0	25.5	25.5
Ethylene glycol	48.3	29.3	19.0	3.0	30.1
Glycerol	64.0	34.0	30.0	3.92	57.4

**Table 8 materials-15-04840-t008:** Volume parameter of CEBM.

Volume Parameter	Emulsified Bitumen Contents (%)		
	8	10	12
Bulk specific gravity	2.594	2.589	2.581
Theoretical maximum density	2.610	2.605	2.599
Air voids (%)	0.61	0.61	0.69

**Table 9 materials-15-04840-t009:** SFE parameters of bitumen and aggregate (25 °C, mJ/m^2^).

Reagent	SFE $\left(\gamma_{L}\right)$	Dispersion Component $\left(\gamma_{L}^{d}\right)$	Polarity Component $\left(\gamma_{L}^{p}\right)$	SFE Acidity Parameter $\left(\gamma_{L}^{+}\right)$	SFE Alkalinity Parameter $\left(\gamma_{L}^{-}\right)$
demulsification bitumen	21.43	18.65	2.78	0.114	17.013
70# bitumen	20.02	19.88	0.14	0.001	6.557
SBS modified bitumen	17.71	13.17	4.54	1.859	2.770
Limestone aggregate	71.52	37.33	34.18	4.036	72.369
Cement block	205.11	125.95	79.16	74.786	20.947

**Table 10 materials-15-04840-t010:** Flowability of CEBM with different cement and emulsified bitumen contents.

Emulsified Bitumen	Flowability of Different Cement Content (s)		
	8%	10%	12%
8% BCR	18.2	19.6	22.1
10% BCR	16.4	18.6	20.2

**Table 11 materials-15-04840-t011:** Low temperature bending test results.

Mixture	Mid-Span Deflection (mm)	Bending Tensile Strength (MPa)	Bending Tensile Strain (με)	Bending Stiffness Modulus (MPa)
CEBM	0.52	4.28	2503.08	1710.21

**Table 12 materials-15-04840-t012:** Marshall stabilities of CEBM with and without water immersion.

Emulsified Bitumen and Cement Content	Marshall Stability (25 °C)		Marshall Stability (60 °C)	
	Unconditioned Group (kN)	Conditioned Group (kN)	Unconditioned Group (kN)	Conditioned Group (kN)
8% BCR + 8% cement	10.12	9.35	8.26	8.35
8% BCR + 10% cement	14.01	15.70	8.71	11.35
8% BCR + 12% cement	13.85	16.52	10.88	12.30
10% BCR + 8% cement	9.56	9.34	8.68	8.54
10% BCR + 10% cement	12.26	14.68	9.93	11.50
10% BCR + 12% cement	13.64	16.62	10.50	11.89

## Data Availability

The data presented in this study are available on request from the corresponding author.

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
