# Peer review of "Research on Design and Performance of Self-Compacting Cement Emulsified Bitumen Mixture (CEBM)"

_materials, 2022, doi:10.3390/ma15144840_

Round 1
Reviewer 1 Report
In this paper, the authors investigated the use of self-compacting cement emulsified bitumen mixture as repair material for potholes in asphalt pavement. The article is interesting and supported by an adequate experimental campaign. Furthermore, I appreciate the effort to correlate the experimental results with the physical and chemical behavior of the material.
In general, I believe that the article is worth for publication.
Here just few comment, mostly related to the drafting:
In 2.2, I suggest you to add a short definition/description of Gussasphalt concrete;
In figure 1, the legend is too general, please detail it;
What about the skid resistance of this material once is used as repair material? Please add some comment;
In 3.3 you have problems with the reference of the figures in the text, please correct it;
In conclusion, what is the optimum in terms of cement content and emulsified bitumen content?
Reviewer 2 Report
The paper presents an experimental study aimed at developing a cement emulsified bituminous mixture for road maintenance. Though several laboratory tests, it was demonstrated to be feasible in terms of technical performance, thus it could be suitable for the addressed aim.
1. Within abstract, it should be preferred to mention the “target” of the designed mixture (maintenance). This because CEBMs are often used for deep pavement layers (this is not the case, and should be described in the abstract).
2. Most figures report error bars, thus it is supposed that various test replicates have been experimentally developed. Please, add in the text the number of executed replicates for each test.
3. When introducing figures in the text, numeration is often wrong (figure’s number seemed to disappear). Please, check and fix all figure’s numeration.
4. The caption of Figure 4 must be revised (“a” and “b” have to be detailed).
5. Citation’s numeration must be revised since it is not sequential (see numbers from 30). Please, do an extensive review to fix all.
6. About Marshall tests: probably, experimental data regarding Marshall flows are available. If inserted in the paper, they could be additional information about the mixes’ mechanical performance (and about “flowability” also).
7. Line 451: “Th” should be “The”.
8. The same consideration addressed to abstract could be repeated for conclusion. Thus, better conclusions should specify the mixture suitability with respect to road works, and the subsequent implications also.
Reviewer 3 Report
The Paper topic " Research on Design and Performance of Self-compacting Cement 2 Emulsified Bitumen Mixture (CEBM)" is under the scope of a materials journal, however, the paper needs major revision before acceptance.
1- Introduction: please try more research papers to show the research gap. The introduction is not strong enough to support the author's claim of novelty. Must improve it accordingly.
line 40: "Due to the increase of the traffic volume and "heavy load"??? why using "---", not necessary.
line 81-103: the authors need to show the study aims!! should use a clear and separate section to show the novelty and research objectives/aims.
2- Abstract: line 20: " A new cement emulsified bitumen mixture (CEBM) with early-strength, self-compacting 20 and room-temperature construction characteristics was innovated and invented" why and how author claims the " innovated "?, clarify.
line 24-25: "The results show that, before the demulsification of emulsified bitumen, the 24 adhesion work between emulsified bitumen and aggregate is negative, that means the adhesion between emulsified asphalt and aggregate will not occur spontaneously", re-write for better presentation.
line 32-33: "This material integrates the hardening effect of cement and the viscoelastic performance of bitumen, and has good workability, mechanical property and road performance" authors need to give the optimum conditions of the design.
References citation though the text and list should follow the journal format.
3. Materials and experimental methods: It is highly recommended to use a flow chart to show the materials and experimental programs including testing methods.
4. figure 5 needs a reference.
5. please do not mix between using bitumen and asphalt. keep the consistency of using the term "bitumen" if referring to the binder.
6. results and discussion of 3.3. Mixture performance test of CEBM, including sections 3.3.1, 3.3.2, and 3.3.3 all sections MUST be improved and show strong discussion in terms of the obtained data and in comparison with the previous research as added in the introduction.
7. Conclusion should have the optimum data of the design that improves the mixture performance.
line 443-445: " Before demulsification, the adhesion work between the emulsified bitumen and aggregate is negative, and the adhesion between them will not occur spontaneously. Meanwhile, the emulsified bitumen exists in the form of liquid, which improves the workability of the CEBM and ensures it can be evenly mixed and self-compacted" ? please re-write for better presentation.
finally, please add future works and recommendations.
English needs major revision.
Round 2
Reviewer 3 Report
Dear authors
Paper needs English language and style are fine/minor spell check required.
And paper accepted in the current version.